# Comparison of the Protective Effects of Bee Venom Extracts with Varying PLA_2_ Compositions in a Mouse Model of Parkinson’s Disease

**DOI:** 10.3390/toxins11060358

**Published:** 2019-06-19

**Authors:** Kyung Hwa Kim, Minhwan Kim, Jaehwan Lee, Hat Nim Jeon, Se Hyun Kim, Hyunsu Bae

**Affiliations:** 1Department of Physiology, College of Korean Medicine, Kyung Hee University, Seoul 02447, Korea; kyungkim@khu.ac.kr (K.H.K.); mcmhkim@hanmail.net (M.K.); moneaven@hanmail.net (J.L.); thdtkfldhk@naver.com (H.N.J.); 2Inist ST Co. Ltd., 159 Sagimakgol-ro, Jungwon-gu, Seongnam-si, Gyeonggi-do 13202, Korea; sean0101@inistst.com

**Keywords:** bee venom, phospholipase A_2_, melittin, Parkinson’s disease, neuroprotection, inflammation, regulatory T cells

## Abstract

Bee venom contains a number of pharmacologically active components, including enzymes and polypeptides such as phospholipase A_2_ (PLA_2_) and melittin, which have been shown to exhibit therapeutic benefits, mainly via attenuation of inflammation, neurotoxicity, and nociception. The individual components of bee venom may manifest distinct biological actions and therapeutic potential. In this study, the potential mechanisms of action of PLA_2_ and melittin, among different compounds purified from honey bee venom, were evaluated against Parkinson’s disease (PD). Notably, bee venom PLA_2_ (bvPLA_2_), but not melittin, exhibited neuroprotective activity against PD in a 1-methyl-4-phenyl-1,2,3,6-tetrahydropyridine (MPTP) mouse model of PD. MPTP-induced behavioral deficits were also abolished after bvPLA_2_ treatment, depending on the PLA_2_ content. Further, bvPLA_2_ administration activated regulatory T cells (Tregs) while inhibiting inflammatory T helper (Th) 1 and Th17 cells in the MPTP mouse model of PD. These results indicate that bvPLA_2_, but not melittin, protected against MPTP and alleviated inflammation in PD. Thus, bvPLA_2_ is a promising and effective therapeutic agent in Parkinson’s disease.

## 1. Introduction

Parkinson’s disease (PD) is an age-related neurodegenerative disorder characterized by the progressive loss of dopaminergic (DA) neurons in the substantia nigra [1]. Generally, most cases of PD are of unclear etiology and occur sporadically. Evidence suggests that inflammation is one of the key contributing factors to the pathogenesis of PD [2]. The enhanced inflammatory response in PD has been detected in the post-mortem brain tissue of PD patients, which demonstrated increased T cell infiltration [3]. Numerous studies have expanded our understanding of the potential role of inflammation in PD pathogenesis. Specifically, recent studies support the role of regulatory T cells (Tregs) in inhibiting T cell inflammatory factors associated with immunosuppressive activity in the brain, which is the potential target in neurodegenerative disease.

Bee venom (BV), delivered by bee sting, consists of a complex mixture of polypeptides, enzymes, lipids, and bioactive amines [4]. BV has been widely associated with significant therapeutic effects under a variety of conditions, including for inflammatory [5] and neurodegenerative diseases [6]. Based on the molecular target, each component of bee venom may have different pharmacological activities. Hence, it is essential to understand the biological function of each component for bee-venom-based drug discovery.

Previously, we reported that bee venom phospholipase A_2_ (bvPLA_2_) protected the brain against neurodegenerative disorders such as Parkinson’s disease [7] and Alzheimer’s disease [8]. Even though the mechanism of action is still unclear, it appears that bvPLA_2_ may have a neuroprotective anti-inflammatory effect, mainly by inducing the activation of regulatory T cells (Tregs). Moreover, there is mounting evidence to support the pharmacological aspects of melittin, one of the main components of honey BV [9]. Indeed, some components of BV, including bvPLA_2_ and melittin, are currently under investigation as promising therapeutic tools. Unfortunately, there are many challenges and issues involved in BV-based drug discovery. Specifically, a clear functional characterization of each BV component needs to be evaluated.

In the present study, we purified bioactive components of BV, such as bvPLA_2_ and melittin, from crude BV by obtaining different concentrations of the components. We investigated the neuroprotective effects of bvPLA_2_ or melittin against Parkinson’s disease in mouse models by administering purified bvPLA_2_ and melittin extracts. We also analyzed the anti-inflammatory effects of purified extracts, based on the activation of microglia and Tregs in the mice.

## 2. Results

### 2.1. Purification PLA_2_ and Melittin from Crude Bee Venom

In order to determine the therapeutic potential of bvPLA_2_ and melittin, we isolated and purified them from the active component of BV. As our primary goal was to identify and compare the therapeutic effects of bvPLA_2_ and melittin, we obtained multiple extracts with diverse compositions of phospholipase A_2_ (PLA_2_) and melittin. The different products were quantified using high-performance liquid chromatography (HPLC) (Figure 1). Based on the corresponding HPLC profiles, the appropriate conditions for separation and purification were established. Furthermore, a satisfactory yield of bvPLA_2_ and melittin was obtained after removing the endotoxins and environmental pollutants such as heavy metals. Thus, the purified extracts appeared to be appropriate for the assessment of the therapeutic potential of bvPLA_2_ and melittin.

### 2.2. BvPLA_2_ with Enriched PLA_2_ Levels Protects against Neurotoxicity in Parkinson’s Disease in MPTP-Induced Mice

First, we carried out a pole test to examine the efficacy of bvPLA_2_ or melittin against motor deficits induced by the neurotoxin, 1-methyl-4-phenyl-1,2,3,6-tetrahydropyridine (MPTP). Mice were treated with each purified extract for six consecutive days, beginning 1 day after MPTP treatment (Figure 2A). As expected, MPTP-injected mice required a longer timeframe to orient downwards (Figure 2B) and a longer period to descend the pole (Figure 2C) when compared with the controls. By contrast, the administration of bvPLA_2_ with enriched PLA_2_ extracts (78% and 98%) effectively suppressed any motor deficits in MPTP-challenged mice. No significant difference was found between the two groups of mice injected with these enriched PLA_2_ agents. By contrast, no significant improvement in motor function was observed in PD mice treated with melittin. Even the MPTP-challenged mice injected with highly purified melittin (99%) showed no significant difference in motor function.

### 2.3. BvPLA_2_ with Enriched PLA_2_ Rescues Loss of Dopaminergic Neurons in MPTP-Treated Mice

We further investigated whether bvPLA_2_ inhibited the loss of dopaminergic neurons in Parkinson’s disease in a PLA_2_-dependent manner. MPTP exposure induced dramatic cellular loss of dopaminergic neurons in the substantia nigra of mice, based on immunohistochemical assays for tyrosine hydroxylase (Figure 3). However, purified bvPLA_2_ protected against MPTP-induced neuronal loss, and the protection was positively correlated with higher PLA_2_ content. Approximately 78% of the purified bvPLA_2_ was adequate to prevent neuronal loss in Parkinson’s disease. Similar to the effects on motor function, melittin appeared to induce no significant changes in the loss of dopaminergic neurons, as demonstrated by their residual population. All these observations suggest that bvPLA_2_-based intervention is an efficient strategy to inhibit neurodegeneration and to rescue neurobehavioral function in Parkinson’s disease.

### 2.4. BvPLA_2_ with Enriched PLA_2_ Levels Induces Differentiation of Regulatory T Cells in MPTP-Challenged Mice

A growing body of evidence has shown that Tregs control the privileged immune status of mouse brains in neurodegenerative disease [10]. Interestingly, a Treg dysfunction was observed in PD patients [11] and in an animal model of PD [12]. Thus, we investigated whether the neuroprotective action of bvPLA_2_ was related to Tregs in Parkinson’s disease. We found no significant difference in CD4^+^CD25^+^Foxp3^+^ Treg cell populations between control and MPTP mice (Figure 4). However, bvPLA_2_ administration induced Treg cell differentiation in mice, depending on the PLA_2_ content. By contrast, melittin treatment did not alter the proportion of Tregs in mice when compared with control mice. These findings suggest that bvPLA_2_, and not melittin, stimulated the differentiation of Treg cells and their immunosuppressive and anti-inflammatory activity.

### 2.5. BvPLA_2_ with Enriched PLA_2_ Suppresses the Differentiation of CD4^+^ Effector T Cells in PD

A number of studies have suggested the importance of maintaining the balance between T cell subsets for immune homeostasis [13]. Thus, we examined whether mouse models of PD displayed an imbalance between CD4+ T cell subtypes. Clearly, MPTP-treated mice exhibited increased numbers of Th1 and Th17 cells, based on the secretion of interferon-γ (IFN-γ) by Th1 cells and interleukin-17A (IL-17A) by Th17 cells (Figure 5). However, bvPLA_2_ treatment reduced the cellular populations of CD4^+^ T cells with Th1 and Th17 phenotypes in MPTP-challenged mice. This inhibitory effect of bvPLA_2_ appeared to show a positive correlation with PLA_2_ content. Conversely, no such inhibitory effects against Th1 or Th17 differentiation were observed in mice treated with melittin in a dose-dependent manner.

## 3. Discussion

In this study, we isolated the bioactive components of PLA_2_ and melittin from crude bee venom. In order to analyze their therapeutic potential, we purified different formulations of PLA_2_ and melittin, exhibiting different purity. We investigated the effect of purified extracts of PLA_2_ and melittin in a mouse model of PD. Interestingly, bvPLA_2_, but not melittin, extracts containing enriched PLA_2_ levels improved motor function and provided substantial protection against MPTP-induced neurodegeneration. Furthermore, we observed an increase in regulatory T cells and a decrease in the inflammatory Th1 and Th17 cells in PD mice, depending on the PLA_2_ concentration.

Indeed, bee venom therapy (BVT) has been widely used to treat several conditions, including rheumatoid arthritis and skin diseases [14]. The most widely studied venoms showing therapeutic effects are those derived from the European honey bee, *Apis mellifera*. These honey BVs contain a variety of biologically active components, including enzymes and peptides [15]. Among these active components, PLA_2_ has been extensively studied for its therapeutic effects in a variety of diseases. Specifically, evidence suggests a therapeutic role of bvPLA_2_ in neurodegenerative diseases [16]. Even though the mechanism of action remains unclear, there is a possible association between PLA_2_ and inflammatory response in the pathogenesis of neurodegenerative disease. Accumulated evidence indicates that enhanced inflammatory response and damage is associated with the development and progression of PD [17]. In particular, Tregs have been of great interest for the treatment of PD due to their anti-inflammatory properties, attenuation of microglial activation, and enhanced neuronal survival in the neurodegeneration of PD [12]. In this study, we observed a clear induction of CD4^+^CD25^+^Foxp3^+^ regulatory T cells, depending on bvPLA_2_. Moreover, PLA_2_ isolated and purified from BV displayed a distinct inhibitory effect on inflammatory Th1 and Th17 phenotypes in a mouse model of PD. Hence, it is possible that increased Tregs induced by bvPLA_2_ suppress inflammation in PD with a concomitant decrease in Th1 and Th17 cell populations. Notably, substantial evidence implicates CD4^+^CD25^+^Foxp3^+^ regulatory T cell dysfunction in the pathogenesis of PD, which suggests that it may be a major contributing factor in the progression of neurodegeneration [10,12]. However, further studies are needed to provide a deeper understanding of the mechanisms underlying the therapeutic effects of bvPLA_2_ in PD.

Melittin, one of the main active components of BV, is a peptide with diverse therapeutic activities, including anti-microbial, anti-tumor, and anti-inflammatory effects [18]. However, recent studies reveal distinct biological actions of melittin in a strict dose-dependent manner. Small doses of melittin induce beneficial anti-inflammatory effects [9], while in high doses, melittin elicits pain and exacerbates the inflammatory response [19]. Numerous studies have reported distinct mechanisms of melittin under different conditions [18]. In this study, we found a limited therapeutic effect of melittin against PD. Interestingly, in this study, we observed that a purified BV extract, containing 78% of PLA_2_ and 15% of melittin, displayed a strong neuroprotective effect and improved motor function in PD. It is possible that the action of bvPLA_2_ is increased by melittin [20]. Further studies are necessary to demonstrate the relationship between bvPLA_2_ and melittin in their therapeutic activity against neurodegenerative disease.

## 4. Conclusions

Bioactive components of BV such as PLA_2_ and melittin are currently available in the market. Despite their widespread availability and use, their formulations are difficult to obtain. In this study, we isolated and produced BV extracts containing different levels of bvPLA_2_ and melittin. Based on further biological experiments, we observed a strong neuroprotective effect, improved motor function, and inhibition of inflammatory T cell phenotypes in a mouse model of PD in a strictly dose-dependent manner with bvPLA_2_, but not melittin. Overall, bvPLA_2_ may be a potential candidate for treating Parkinson’s disease and other diseases associated with neuroinflammation. 

## 5. Materials and Methods

### 5.1. Preparation and Manufacturing of bvPLA_2_ and Melittin

The standardized BV phospholipase A_2_ was prepared by Inist St Co. Ltd. (Eumseong-gun, South Korea). Briefly, crude BV was purchased from Bee Venom Lab LLC (Tbilisi, GA, USA) and applied to a polytetrafluoroethylene (PTFF) membrane filter (pore size 0.45 μm; Sigma-Aldrich, St. Louis, MO, USA). In order to reduce the volume, the mixtures were concentrated by Ultracel-10 kDa membrane with Pellicon 3 devices (Merck Millipore, Billerica, MA, USA). The separation was carried out using reversed-phase high-performance liquid chromatography (RP-HPLC) on a C18 column (Sigma-Aldrich, St. Louis, MO, USA). The area of the detected peak was measured to determine the recovery of bvPLA_2_ and melittin. Commercial standard bvPLA_2_ (Sigma-Aldrich, St. Louis, MO, USA) was used as PLA_2_ with high purity (98% of PLA_2_). All these procedures were performed in an aseptic good manufacturing practice (GMP) facility. For quality management, a purity test was performed to confirm the absence of detectable heavy metals, endotoxins, or microbes.

### 5.2. Animals

All animal experiments were carried out in accordance with the approved animal protocols and guidelines established by the Kyung Hee Univesity [KHUASP(SE)17-149]. The experiments were conducted on 7 to 8 weeks old make C57BL/6J mice (20–22 g; Japan SLC, Hamatsu, Japan). Mice were maintained in a pathogen-free facility on a 12 h light/dark cycle with food and water provided ad libitum.

### 5.3. MPTP-Induced Mouse Model

MPTP (20 mg/kg; Sigma-Aldrich, St. Louis, MO, USA) was intraperitoneally (i.p.) administered to mice four times a day at 2 h intervals, as previously described [21]. The physical condition of the mice was monitored.

### 5.4. BvPLA_2_ Treatment

MPTP-treated mice were exposed to either bvPLA_2_ or phosphate buffered saline (PBS) for a consecutive 6 days, beginning 1 day after the last MPTP injection. Purified standard bvPLA_2_ formulations and commercial grade bvPLA_2_ (Sigma-Aldrich, St. Louis, MO, USA) formulations were dissolved in PBS and administered as a single daily subcutaneous injection at a concentration of 0.5 mg/kg.

### 5.5. Immunohistochemistry

Immunohistochemical analysis was performed, as previously described [7]. Mouse brain tissues were prepared for histology in 4% paraformaldehyde overnight, washed in PBS, and then immersed in 30% sucrose until they sank. The brains were then sliced into 30 μm coronal sections using a sliding microtome. Next, the sections were processed for 30 min in 1% H_2_O_2_ in PBS and then blocked with PBS buffer containing 1% BSA and 0.1% Triton X-100 during 1 h. After an overnight incubation with tyrosine hydroxylase (TH) antibody (Wako Pure Chemic Industries, Osaka, Japan) at 4 °C, the tissues were incubated with biotinylated goat anti-rabbit immunoglobulin G (IgG) secondary antibody (Vector Laboratories, Burlingame, CA, USA). Following another wash, the brains were subjected to incubation in avidin–biotin complex using ABC kit (Vectastain Elite ABC kit; Vector Laboratories, Burlingame, CA, USA). The chromogen reactions were then performed with diaminobenzidine (DAB) (Vector Laboratories, Burlingame, CA, USA).

### 5.6. Unbiased Stereological Estimation

An unbiased stereological estimation of the number of TH-positive cells was performed using an optical fractionator, as previously described with minor modifications [22]. Briefly, the sections extending from the rostral tip of the substantia nigra pars compacta (SNpc) to the caudal end of the substantia nigra pars reticulate (SNR) were selected and counted using Olympus CAST-Grid system (Olympus). The total number of cells was calculated using the optical fractionator.

### 5.7. Flow Cytometry

To determine the number of regulatory T cells in the splenocytes, flow cytometry was performed, as previously described [7]. The splenocytes were washed with PBS and stained with fluorescein isothiocyanate (FITC)-conjugated anti-mouse CD4 (eBioscience, San Diego, CA, USA) and phycoerythrin (PE)-conjugated anti-mouse CD25 (eBioscience, San Diego, CA, USA). Next, the cells were fixed and stained with Alexa Fluor 647 anti-mouse Foxp3 (BD Biosciences, San Jose, CA, USA) overnight at 4 °C in the dark. After washing, the cells were analyzed. In order to measure Th1 and Th17 cells in the splenocytes, FITC-conjugated anti-mouse CD4 (eBioscience, San Diego, CA, USA), PE-conjugated anti-mouse IFN-γ (eBioscience, San Diego, CA, USA), and PerCP/Cyanine5.5-conjugated anti-mouse IL-17A (eBioscience, San Diego, CA, USA) were stained according to the supplier’s instructions.

### 5.8. Behavioral Test

The pole test was performed to measure motor coordination and balance, as previously described [23]. In brief, the mice were placed on the top of a gauze-banded wooden pole (diameter 1 cm; height 50 cm). The time at which mice successfully oriented themselves downward and climbed down was measured using a cut-off limit of 30 s. The average of three trials was used. Trials were excluded if the mouse dropped or jumped off.

### 5.9. Statistical Analysis

Statistical analysis was carried out using GraphPad Prism software (v5.0; GraphPad Software, La Jolla, CA, USA). Conditions were compared using a one-way analysis of variance (ANOVA) followed by Newman–Keuls test for multiple comparisons. A paired *t*-test was used to compare two groups. Data were expressed as the mean ± standard error of the mean (SEM). *p* < 0.05 was considered significant.

## Figures and Tables

**Figure 1 toxins-11-00358-f001:**
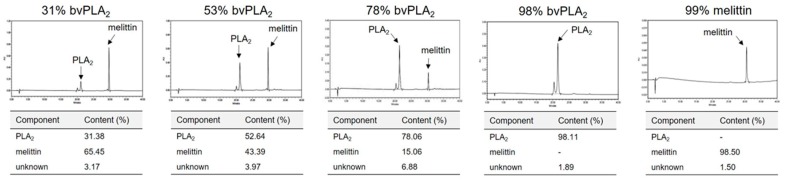
Purification and manufacture of bee venom extracts containing PLA_2_ and melittin. Bioactive compounds from bee venom were purified according to their PLA_2_ and melittin content using high-performance liquid chromatography (HPLC) (**upper panel**). The components of the purified extracts derived from bee venom are displayed (**lower panel**).

**Figure 2 toxins-11-00358-f002:**
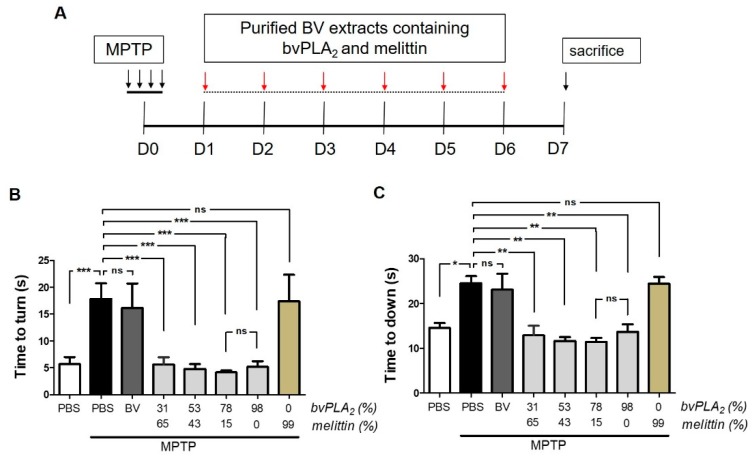
Beneficial effects of bvPLA_2_ enriched with PLA_2_ against motor deficits in the neurotoxin 1-methyl-4-phenyl-1,2,3,6-tetrahydropyridine (MPTP)-challenged mice. (**A**) Experimental timeline: the bee venom (BV) extracts containing different levels of PLA_2_ and melittin were administered to MPTP-exposed mice for 6 consecutive days, starting on day 1 after MPTP injection. (**B**,**C**) The motor deficits induced by MPTP were examined on Day 6 after MPTP. Time to orient downward and descend the pole was measured with the pole test. The data are expressed as the means ± standard error of the mean (SEM). n = 3–4 per group; **p* < 0.05, ns, not significant.

**Figure 3 toxins-11-00358-f003:**
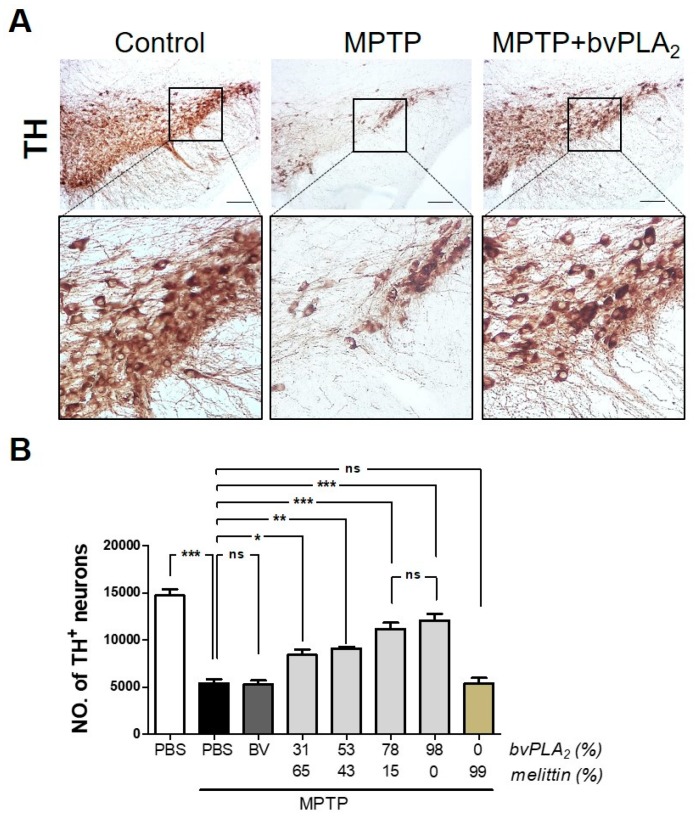
Neuroprotective effect of standard bvPLA_2_ with enriched PLA_2_ on dopaminergic neurons in MPTP-injected mice. On Day 7 post-MPTP, immunohistochemical analysis of tyrosine hydroxylase (TH) was performed in MPTP mice injected with bee venom extracts containing PLA_2_ and melittin. (**A**) Representative images of sections containing TH-positive neurons, with high magnifications of dotted squares in substantia nigra pars compacta (SNpc). Phosphate buffered saline (PBS)-injected group was assigned to control. Upper panels are low magnification images (10×). Lower panels are high magnification images of the boxed areas in corresponding upper panels. (**B**) Unbiased stereological estimation for TH-positive neurons in substantia nigra. MPTP + bvPLA_2_ group: MPTP-exposed mice were treated with the enriched bvPLA_2_ (78% of PLA_2_ and 15% of melittin). Data are expressed as the means ± standard error of the mean (SEM). n = 3–4 per group; **p* < 0.05, ***p* < 0.01, ****p* < 0.001, ns, not significant. Scale bar: 100 μm.

**Figure 4 toxins-11-00358-f004:**
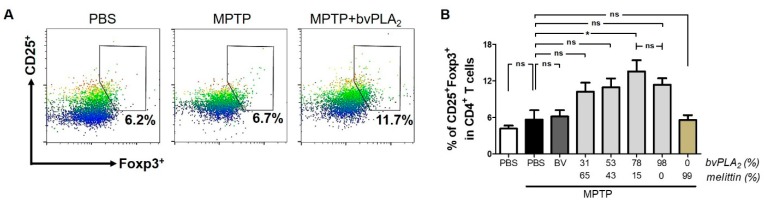
Induction of regulatory T cells in MPTP-treated mice following administration of bvPLA_2_. Seven days after MPTP injection, CD4^+^CD25^+^Foxp3^+^ regulatory T cells from splenocytes of each group mice were analyzed by flow cytometry. Bee venom agents with different levels of PLA_2_ and melittin were administered to MPTP-challenged mice. Representative plots (**A**) and quantification (**B**) of CD25^+^Foxp3^+^ cells in CD4^+^ T cells are shown. MPTP + bvPLA_2_ group: MPTP-exposed mice were treated with enriched bvPLA_2_ (78% of PLA_2_ and 15% of melittin). Data are expressed as the means ± standard error of the mean (SEM). n = 3–4 per group; **p* < 0.05, ns, not significant.

**Figure 5 toxins-11-00358-f005:**
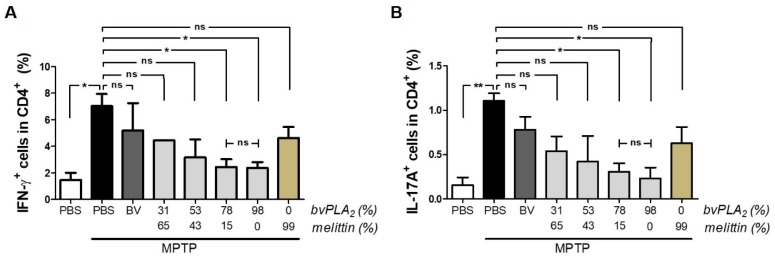
Inhibitory effect of bvPLA_2_ on the differentiation of CD4^+^ effector T cells in MPTP-treated mice. IFN-γ- (**A**) and IL-17A-positive (**B**) splenic CD4^+^ T cells were analyzed using flow cytometry on Day 7 post-MPTP to evaluate the number of Th1 and Th17 cells, respectively. Bee venom extracts containing different levels of PLA_2_ and melittin were administered to MPTP-treated mice. Data are expressed as the means ± standard error of the mean (SEM). n = 3–4 per group; **p* < 0.05, ***p* < 0.01, ns, not significant.

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
