# Peer review of "Comparison of the Protective Effects of Bee Venom Extracts with Varying PLA2 Compositions in a Mouse Model of Parkinson’s Disease"

_toxins, 2019, doi:10.3390/toxins11060358_

Round 1

Reviewer 1 Report

Reviewer’s report

Manuscript Comparison of the protective effects of bee venom extracts with varying PLA2 compositions in a mouse model of Parkinson’s disease.

Beneficial effects of treatment with bee venom phospholipase A2 were observed in an experimental mouse model of Parkinson’s disease. The topic is important since new effective treatment modalities are needed for the disease.

In the current study, a PD-like condition was induced experimentally in mice by administration of 1-methyl-4-phenyl-1,2,3,6-tetrahydropyridine (MPTP) that is known to damage dopamine cells in substantia nigra. PLA2 was harvested from honey bee venom by HPLC and injected in the MTPT-treated animals. The motor performance and the brain cell injury of the experimental animals were examined by a pole test and microscopy, respectively.

The investigation employed well known methods in a reliable manner. The results largely confirm earlier observations on the beneficial effect of bee venom PLA2 in Parkinson’s disease, e.g. Ye et al. Experimental Molecular Medicine 2016, 48, e244; doi:10.1038/emm.2016.49. The mechanism of action of this treatment modality remained unknown in the current study, although immunological mechanisms were implied as proposed earlier, e.g. Yan et al. Molecular Medicine Reports 2014, 10, 2223; Chung et al. Journal of Immunology 2015, 195, 4853.

I have the following queries:

Line 235. It might serve the reader if the pole test is described by a couple of words here.

Line 213. Immunohistochemistry. Tissue fixation (fixative; mode of administration) should be explained here).

Line 216. What is anti-secondary antibody?

Line 101. Figure 3. Please explain the Control

Line 101. Figure 3. What are the magnifications of the micrographs?

Author Response

Response to Reviewer 1 Comments

We appreciate the reviewer for his/her detailed comments and suggestions for our manuscript. After completion of the suggested edits, we believe that the revised manuscript has benefitted from in an improvement. Thank you so much for the effort you made.

Below, you will find a point by point description of how each comment was addressed in the manuscript. Original reviewer comments in black, responses in red.

Point 1:

Line 235. It might serve the reader if the pole test is described by a couple of words here.

Response 1:

In response to reviewer’s comment, we have rewritten each “Methods section” with more methods. Also, we switched the subtitle from ‘Pole test’ to ‘Behavioral test’ to clearly describe this test.

(Line 248-251)

5.8. Behavioral test

The pole test was performed to measure motor coordination and balance as previously described23. In brief, the mice were placed on the top of a gauze-banded wooden pole (diameter 1 cm; height 50 cm).

Point 2:

Line 216. What is anti-secondary antibody?

Response 2:

In response to this comment, we have modified and added details in ‘Immunohistochemistry’ Methods section. As secondary antibody, we used biotinylated goat anti-rabbit IgG secondary antibody.

(Line 227-232)

After an overnight incubation with primary TH antibody (Wako Pure Chemic Industries, Osaka, Japan) at 4°C, the tissues were incubated with biotinylated goat anti-rabbit IgG secondary antibody (Vector Laboratories, Burlingame, CA, USA). Following another wash, the brains were subjected to incubation in avidin-biotin complex using ABC kit (Vectastain Elite ABC kit; Vector Laboratories, Burlingame, CA, USA). Then, the chromogen reactions were performed with diaminobenzidine (DAB) (Vector Laboratories, Burlingame, CA, USA).

Point 3:

Line 213. Immunohistochemistry. Tissue fixation (fixative; mode of administration) should be explained here).

Response 3:

As the reviewer suggested, we have rewritten‘Immunohistochemistry’ Methods section with including fixative, mode of administration and sample preparation for staining).

(Line 223-227)

Mouse brain tissues were prepared for histology in 4% paraformaldehyde overnight, washed in PBS, and then immersed in 30% sucrose until they sank. Then the brains were sliced into 30 μm coronal sections using a sliding microtome. Then the sections were processed for 30 min in 1% H2O2 in PBS, blocked by 1 h with 1% BSA, 0.1% Triton X-100 in PBS.

Point 4:

Line 101. Figure 3. Please explain the Control

Response 4:

Thank you for your thoughtful review. We explained the ‘Control’ in Figure 3 legend.

(Line 105)

PBS-injected group was assigned to control.”

Point 5:

Line 101. Figure 3. What are the magnifications of the micrographs?

Response 5:

Thank you so much for pointing out. We obtained the original images at 10x magnification.

We have added information in Figure 3 legend.

(Line 105-107)

“Upper panels are low magnification images (x10). Lower panels are high magnification images of the boxed areas in corresponding upper panels.”

Reviewer 2 Report

The description of the strain and gender of the animals should be included as well as the post hoc test used.

Author Response

Response to Reviewer 2 Comments

We would like to thank the reviewer for careful and thorough reading of this manuscript. Thank you so much for the opportunity to revise our paper. We appreciate the effort you made to improve this manuscript and are grateful for your insightful comments. Below, you will find a point by point description of how each comment was addressed in the manuscript. Original reviewer comments in black, responses in red.

Point 1:

Question: Are the methods adequately described?

(Answer: Can be improved)

Response 1:

In response to reviewer’s comment, we have modified “Methods section” to be much more detailed to provide much information. We rewritten each “Methods section” with more methods. Moreover, we also generated highlighted version of revised manuscript which the changes in the main text were highlighted.

Point 2:

Question: English language and style

(Answer: English language and style are fine/minor spell check required)

Response 2:

We highly appreciate for the reviewer’s careful reading of our manuscript. We have carefully checked spelling to eliminate grammatical errors. The changes in the main text are highlighted in the revised manuscript.

Point 3:

The description of the strain and gender of the animals should be included as well as the post hoc test used.

Response 3:

Thank you for pointing out.

Firstly, we generated ‘Animals’ part in Methods section to describe information about the animals which were used in this study as well as about care and handling animals for this study.

(Line 206-211)

5.2. Animals

All animal experiments were carried out in accordance with the approved animal protocols and guidelines established by the Kyung Hee Univesity [KHUASP(SE)17-149]. The experiments were conducted on 7- to 8-wk-old make C57BL/6J mice (20-22 g; Japan SLC, Hamatsu, Japan). Mice were maintained in a pathogen-free facility on a 12 h light/dark cycle with food and water provided ad libitum.”

Moreover, we rewritten ‘Statistical analysis’ in Method sections with adding information including the details of multiple comparison test.

(Line 254-258)

5.9. Statistical analysis

Statistical analysis was carried out using GraphPad Prism software (v5.0; GraphPad). Conditions were compared using a one-way ANOVA followed by Newman-Keuls test for multiple comparisons. A paired t-test was used to compare two groups. Data were expressed as the mean ± SEM. P < 0.05 was considered significant.
